# Prevalence of Gastrointestinal Symptoms in Chinese Community-Dwelling Adults with and without Diabetes

**DOI:** 10.3390/nu14173506

**Published:** 2022-08-26

**Authors:** Miaomiao Sang, Tongzhi Wu, Xiaoying Zhou, Michael Horowitz, Karen L. Jones, Shanhu Qiu, Haijian Guo, Bei Wang, Donglei Wang, Christopher K. Rayner, Zilin Sun

**Affiliations:** 1Department of Endocrinology, Zhongda Hospital, Institute of Diabetes, School of Medicine, Southeast University, Nanjing 210009, China; 2Centre of Research Excellence in Translating Nutritional Science to Good Health, Adelaide Medical School, The University of Adelaide, Adelaide, SA 5000, Australia; 3Endocrine and Metabolic Unit, Royal Adelaide Hospital, Adelaide, SA 5000, Australia; 4Department of General Practice, Zhongda Hospital, Institute of Diabetes, School of Medicine, Southeast University, Nanjing 210009, China; 5Department of Integrated Services, Jiangsu Provincial Center for Disease Control and Prevention, Nanjing 210009, China; 6Key Laboratory of Environmental Medicine and Engineering Ministry of Education, Department of Epidemiology and Health Statistics, School of Public Health, Southeast University, Nanjing 210009, China; 7Department of Gastroenterology and Hepatology, Royal Adelaide Hospital, Adelaide, SA 5000, Australia

**Keywords:** gastrointestinal symptoms, diabetes, Diabetes Bowel Symptom Questionnaire

## Abstract

Background: Gastrointestinal symptoms have been reported to occur frequently in diabetes, but their prevalence in Chinese community-dwelling individuals with diabetes is unknown. The present study aimed to address this issue and explore the risk factors for gastrointestinal symptoms. Methods: A total of 1304 community-dwelling participants (214 with diabetes, 360 with prediabetes and 730 with normoglycemia) were surveyed for gastrointestinal symptoms using the Diabetes Bowel Symptom Questionnaire. Logistic regression analyses were applied to identify risk factors for gastrointestinal symptoms. Results: Of the overall study population, 18.6% reported at least one gastrointestinal symptom, without a significant difference between subjects with normoglycemia (17.7%), prediabetes (19.7%) and diabetes (20.1%). In all three groups, lower gastrointestinal symptoms, particularly diarrhea and constipation, were the most frequent. There was an interaction between age (≥65 years) and diabetes on the prevalence of at least one gastrointestinal symptom (*p* = 0.01) and of constipation (*p* = 0.004), with these being most frequent in subjects with diabetes aged ≥ 65 years. After multivariable adjustment, female gender and older age were associated with increased odds of at least one gastrointestinal symptom, specifically lower gastrointestinal symptoms. Older age was also associated with an increase in upper gastrointestinal symptoms. Conclusions: Gastrointestinal symptoms are common in Chinese community-dwelling adults with and without diabetes. Females, and the elderly with diabetes, are at an increased risk of symptoms.

## 1. Introduction

Gastrointestinal (GI) symptoms occur frequently in the community and are associated with a substantial increase in health costs and a reduction in quality of life [1]. The pathogenesis of GI symptoms is complex and may be related to disturbances of gastrointestinal motility, visceral sensitivity, mucosal and immune function, the gut microbiota and/or central nervous system processing [2]. Factors associated with an increased risk of GI symptoms include female gender [1], older age [1], psychological comorbidities (e.g., anxiety and depression) [3] and obesity [4]. 

Many studies from western countries have reported that individuals with diabetes have a higher prevalence of upper and lower GI symptoms than controls [3], with risk factors including the duration of diabetes, presence of diabetic complications (e.g., peripheral neuropathy) and poor glycemic control [5,6,7,8]. Moreover, GI symptoms can occur as an adverse effect of a number of glucose-lowering drugs, which may compromise compliance with therapy [9]. An understanding of the prevalence and the determinants of GI symptoms in diabetes is of relevance to the development of effective strategies for management. 

There are substantial variations in the reported prevalence of GI symptoms in diabetes between studies, probably reflecting differences in the characteristics of the study population and the method used to assess GI symptoms. In a cohort of outpatients with diabetes in Texas (*n* = 136), self-reported GI symptoms occurred in 76% of patients [10]. In another case-control study, the GI symptoms of Korean outpatients with and without type 2 diabetes (T2D) (*n* = 190 per group) were assessed using a questionnaire based on the Rome II criteria for functional GI disorders [7]. Compared to those without diabetes, T2D patients were reported to have similar rates of at least one GI symptom (72% vs. 62%) and of lower GI symptoms (58% vs. 55%), but a higher rate of upper GI symptoms (43% vs. 31%) [7]. By contrast, a US community-based study (*n* = 743) using the Bowel Disease Questionnaire (BDQ) showed that the prevalence of GI symptoms was similar between T2D patients and healthy controls [11]. The Diabetes Bowel Symptom Questionnaire (DBSQ) is the first validated GI questionnaire designed specifically for the comprehensive assessment of GI symptoms in diabetes [12]. The application of the DBSQ in a community-dwelling population in Australia (>8600 responses) revealed that 18.2% of participants with diabetes had upper GI symptoms and 26.0% had lower GI symptoms, compared to 15.3% and 18.9%, respectively, in non-diabetic controls [5]. 

Until now, there has been limited information concerning the prevalence of GI symptoms in Chinese patients with diabetes. About 70% of T2D outpatients in Hong Kong (*n* = 149) reported GI symptoms when studied about two decades ago using a questionnaire developed by Horowitz et al. [13]; both upper and lower GI symptoms were more prevalent than in community-based controls [14]. However, patients attending hospital outpatient clinics are unlikely to be typical of people with T2D in the general community, and we are unaware of any community-based studies of GI symptoms in Chinese populations. Although a recent global epidemiological study of functional GI disorders reported that GI symptoms are relatively common in community-dwelling Chinese adults, whether assessed via interview (22.7%) or internet survey (34.4%) [1], this study did not evaluate the prevalence of GI symptoms specifically in individuals with diabetes. 

The current study aimed to determine the prevalence of GI symptoms in Chinese community-dwelling adults with and without diabetes using the DBSQ and to explore risk factors for GI symptoms. 

## 2. Materials and Methods

### 2.1. Participants

Participants in our study were recruited from the Study on Evaluation of iNnovative Screening tools and determInation of optimal diagnostic cut-off points for type 2 diaBetes in Chinese muLti-Ethnic (SENSIBLE) [15] and SENSIBLE-Addition studies [16], which were designed to determine the optimal cut-off values of advanced glycation end-products and glycated hemoglobin (HbA1c) for the diagnosis of T2D in China. In these studies, an age- and sex-stratified, random sample of 17,629 community adult dwellers (aged > 18 years) who had lived in their current residence for ≥5 years were recruited from eight provinces of China between 1 November 2016 and 31 July 2017 and were followed up at intervals of ~1.5 years until January 2021 [17]. The current study randomly enrolled a subset of participants (1713 out of a total of 4040 participants) in the Jiangsu, Hebei and Jiangxi provinces, who were surveyed between April 2020 and January 2021. Key exclusion criteria were: (1) a history of GI disease (including chronic gastritis, peptic ulcer and biliary tract disorders); (2) a history of GI surgery (except for appendicectomy); (3) known chronic liver or kidney disease, or heart failure); (4) any history of malignancy. After excluding those who met one or more of these criteria, a total of 1304 individuals remained in the final analysis (Figure 1).

The study protocol was approved by the Human Research Ethics Committee of Zhongda Hospital, Southeast University, Nanjing, China. Written, informed consent was obtained from all participants. 

### 2.2. Measurements

Standardized questionnaires were used to collect demographic data (including age, gender and ethnicity), information regarding health behaviors (including history of smoking and alcohol intake) and medical history (including hypertension, dyslipidemia, diabetes, cardiovascular disease and medication use) by trained interviewers. Anthropometric parameters including height, weight, waist circumference and systolic and diastolic blood pressure (SBP and DBP) were measured according to standard protocols, and body mass index (BMI) was calculated. Smoking status was classified as never (<100 cigarettes in a lifetime), former (>100 cigarettes, but none in the past 30 days) or current (smoked in the previous 30 days). Alcohol intake was classified as none, <10 g per week, 10–100 g per week, >100 g but <300 g per week and ≥300 g per week.

A venous blood sample was drawn after 8 h of fasting for measurement of fasting plasma glucose (FPG), HbA1c, total cholesterol (TC), triglycerides (TG), high-density lipoprotein (HDL), low-density lipoprotein (LDL) and serum creatinine (Cr). An additional venous blood sample was obtained at 120 min after a 75 g oral glucose load in participants without a prior history of diabetes, for the measurement of 2 h plasma glucose (2hPG). FPG, 2hPG, TC, TG, HDL, LDL and Cr levels were measured using an automated chemistry analyzer (Synchron LX-20, Beckman Coulter Inc., Fullerton Pasadena, CA, USA). HbA1c was measured by high-performance liquid chromatography (HPLC; D-10™ Hemoglobin Analyzer, Bio-Rad Inc., Hercules, CA, USA) [15]. Participants with diabetes were also screened for diabetic nephropathy by the measurement of the albumin-to-creatinine ratio (UACR) and diabetic retinopathy (DR) by fundus photography.

### 2.3. Definitions of Glycemic Status and Diabetic Complications

Glycemic status was defined according to the World Health Organization (WHO) 1999 criteria, as: (1) diabetes (FPG ≥ 7 mmol/L, 2hPG ≥ 11.1 mmol/L, current use of hypoglycemic drugs or self-reported history of diabetes), (2) prediabetes, including impaired fasting glucose (6.1 mmol/L ≤ FPG < 7.0 mmol/L and 2hPG < 7.8 mmol/L) and/or impaired glucose tolerance (FPG < 7.0 mmol/L and 7.8 mmol/L ≤ 2hPG < 11.1 mmol/L) and (3) normoglycemia (FPG < 6.1 mmol/L and 2hPG < 7.8 mmol/L).

Albuminuria was defined as UACR ≥ 30 mg/g and categorized into microalbuminuria (30–300 mg/g) or macroalbuminuria (>300 mg/g). DR, defined according to the Early Treatment Diabetic Retinopathy Study (ETDRS) criteria, was categorized into mild non-proliferative (level 20, 31), moderate non-proliferative (level 41), severe non-proliferative (level 51) and proliferative retinopathy (level ≥ 60) [18].

### 2.4. Assessment of GI Symptoms

The DBSQ is a validated questionnaire which asks participants to recall the occurrence of a range of upper and lower GI symptoms within the last 3 months [12]. Symptom terminology in the DBSQ is based primarily on Rome II criteria [19] and allows symptoms/symptom clusters of abdominal pain, irritable bowel syndrome (IBS), ulcer-like dyspepsia, early satiety, postprandial fullness, nausea, vomiting, retching, loss of appetite, abdominal fullness or bloating, gastroesophageal reflux, dysphagia, diarrhea, constipation and fecal incontinence to be elicited. Symptom complexes were assessed as being present if any of the component symptoms occurred more than a quarter of the time, within the last 3 months [12]. For the assessment of GI symptoms in the current study, the DBSQ was translated into Chinese and checked for face validity by a bilingual gastroenterologist. Concurrent validity of the translated questionnaire was evaluated in a sample of 23 inpatients with T2D who completed the questionnaire by face-to-face interview with a study interviewer, followed one week later by a telephone interview (due to restrictions of COVID-19) by an endocrinologist, which yielded acceptable agreement for all questions (kappa values 0.7–1.0, *p* < 0.01).

### 2.5. Statistical Analysis

Data are presented as means ± standard deviations (SDs) or medians (25th percentile, 75th percentile) for continuous variables and as percentages for categorical variables. Differences in parameters across the three groups were compared using one-way analysis of variance and the Kruskal–Wallis test for parametric and non-parametric distributions, respectively and the Chi-squared test, with subgroup comparisons adjusted by Bonferroni’s correction when appropriate. Logistic regression analyses were used to compare the prevalence of GI symptoms between the groups after adjusting for covariates (including age, gender, smoking status, BMI and SBP) and to determine potential factors associated with GI symptoms (including age, gender, BMI, FPG and HbA1c). Interaction was assessed through the incorporation of an interaction term into the logistic regression models. Since a subset of subjects with diabetes were taking oral glucose-lowering drugs, with or without insulin, or had missing information regarding glucose-lowering therapies, which might have affected the prevalence of GI symptoms, a sensitivity analysis was performed that excluded these subjects. Clinical features of individuals with diabetes who reported GI symptoms and those who reported no GI symptoms were also compared to identify potential risk factors associated with GI symptoms in diabetes. Individuals with diabetes were further stratified according to their glucose-lowering strategies and HbA1c to examine differences in the prevalence of GI symptoms. Statistical analyses were performed using SPSS 25.0 (IBM SPSS Statistics for Windows, Version 25.0, IBM Corp., Armonk, NY, USA), with *p* < 0.05 being considered statistically significant.

## 3. Results

### 3.1. Characteristics of the Participants

A total of 1304 participants with complete data was included in the final analysis and grouped as normoglycemia (*n* = 730), prediabetes (*n* = 360) and diabetes (*n* = 214). The clinical characteristics of the enrolled participants are summarized in Table 1. Subjects with prediabetes and diabetes were slightly older and had higher waist circumference, SBP, DBP, FPG, 2hPG, HbA1c, TC and TG, compared to those with normoglycemia. In addition, heart rate was greater in the subjects with diabetes than those with normoglycemia. There were no differences in smoking and alcohol intake between the groups.

Subjects with diabetes included those with a prior history of diabetes (*n* = 91) and those diagnosed in the current study (*n* = 123). Of the 91 subjects with known diabetes, the mean duration of diabetes was 6.5 ± 5.4 years, and 50.5% were treated by oral hypoglycemic drugs, 13.2% by insulin alone or in combination with oral drugs, 19.8% by lifestyle measures only and the rest unclassified due to missing data. Albuminuria and DR were noted in 34.9% and 10.8% of patients with diabetes, respectively, but the majority of these complications was at an early stage (85.9% as microalbuminuria and 72.7% as mild non-proliferative DR).

### 3.2. GI Symptoms in Participants with Normoglycemia, Prediabetes and Diabetes

As shown in Table 2, at least one GI symptom was present in 18.6% of the overall population, and the overall prevalence did not differ significantly between subjects with normoglycemia (17.7%), prediabetes (19.7%) and diabetes (20.1%). Lower GI symptoms, particularly diarrhea and constipation, were the most common in all three groups. The prevalence of IBS did differ between the three groups (*p* = 0.025), being higher in those with prediabetes compared to normoglycemia (3.9% vs. 1.4%, *p* = 0.008). This difference remained significant after the adjustment for gender and age [OR 2.72, 95% CI (1.18, 6.26), *p* = 0.02]. A sensitivity analysis which excluded people with diabetes who were taking oral glucose-lowering drugs, with or without insulin, or had missing information regarding glucose-lowering therapies, showed that the outcomes were essentially unchanged (Appendix A). The prevalence of IBS remained higher in subjects with prediabetes vs. normoglycemia, both with and without the adjustment for gender and age [*p* = 0.008 before adjustment; OR 2.77, 95% CI (1.20, 6.37), *p* = 0.02 after adjustment]. There were no differences in other GI symptoms between the groups.

Subjects in each group were further stratified according to gender (male vs. female) (Table 3) and age (<65 years vs. ≥65 years) (Table 4). As shown in Table 3, female subjects reported a higher prevalence of early satiety (*p* = 0.02), loss of appetite ( *p* = 0.03) and constipation ( *p* = 0.004) in the overall study population. The prevalence of abdominal fullness or bloating (*p* = 0.008) and constipation (*p* = 0.005) was also higher in females in the normoglycemic group. Given that the demographic variables in male and female subjects across the three groups were well-matched, except that the rate of current smoking was higher in males (data not shown), the comparisons in the rate of GI symptoms between male and female subjects were adjusted further for smoking status. In this model, the rate of constipation was shown to be higher in females than males in the overall population [OR 2.02, 95% CI (1.05, 3.89), *p* = 0.036] and in subjects with normoglycemia [OR 3.19, 95% CI (1.08, 9.39), *p* = 0.035].

As shown in Table 4, there was a significant interaction between age (≥65 years vs. <65 years) and the presence of diabetes on the prevalence of at least one GI symptom (*p* for interaction = 0.01), in particular for constipation (*p* for interaction = 0.004). Subjects with diabetes aged ≥ 65 years had a higher rate of at least one GI symptom (34.8% vs. 13.7%, *p* = 0.007) and of constipation (17.4% vs. 2.7%, *p* = 0.005), compared to those with normoglycemia. These differences were evident after the adjustment for age and gender [OR 3.28, 95% CI (1.32, 8.17), *p* = 0.011 for at least one GI symptom; OR 7.32, 95% CI (1.41, 38.07), *p* = 0.018 for constipation]. In addition, the prevalence of at least one GI symptom (16.1% vs. 34.8%, *p* = 0.005), constipation (2.4% vs. 17.4%, *p* < 0.001) and lower GI symptoms (21.7% vs. 9.5%, *p* = 0.025) was all higher in subjects with diabetes aged ≥ 65 years than in those aged < 65 years. The above comparisons were further adjusted for differences in BMI and SBP, which did not affect the outcomes (All *p* < 0.01).

In subjects with diabetes, there were no differences in FPG, HbA1c, the duration of known diabetes or the prevalence of comorbidities between those with and without GI symptoms (Appendix A). There were also no differences in GI symptoms between subsets of subjects with diabetes (i.e., subjects receiving different type of glucose-lowering therapies; subjects with different HbA1c levels) (Appendix A).

### 3.3. Potential Factors Associated with GI Symptoms

As shown in Table 5, after multivariable adjustment, female gender and older age were associated with increased odds of at least one GI symptom [OR 1.36, 95% CI (1.01, 1.84), *p* = 0.04 and OR 1.04, 95% CI (1.02, 1.05), *p* < 0.001, respectively] and of lower GI symptoms [OR 1.44, 95% CI (1.003, 2.08), *p* = 0.048 and OR 1.03, 95% CI (1.01, 1.05), *p* = 0.005, respectively]. Older age was also associated with increased odds of upper GI symptoms [OR 1.04, 95% CI (1.01, 1.06), *p* = 0.004]. However, no significant association of FPG or HbA1c with GI symptoms was observed.

## 4. Discussion

We observed that GI symptoms were common in Chinese community-dwelling adults, occurring in almost 20% of individuals overall, regardless of whether they were classified as having normoglycemia, prediabetes or diabetes. Lower GI symptoms (i.e., diarrhea and constipation) occurred most frequently. Older age and female gender were associated with a higher prevalence of GI symptoms, while individuals with diabetes aged ≥ 65 years had a higher prevalence of at least one GI symptom, and of constipation in particular, when compared to those with normoglycemia.

The overall prevalence of at least one GI symptom in our study is comparable to what was reported in a recent global epidemiological study of functional GI disorders [1], in which 20.7% of the worldwide population and 22.7% of the Chinese community cohort (n = 2710) reported symptoms consistent with at least one functional GI disorder, when symptoms were ascertained by interview. Similar to the global epidemiological study, we showed that older age was associated with increased odds of both upper and lower GI symptoms through interviews. Although a new onset of GI symptoms in the elderly should always prompt consideration of specific underlying causes (e.g., malignancy), it should be recognized that older individuals can also suffer from functional GI disorders [20]. Moreover, we found that females had a higher prevalence of early satiety, loss of appetite and constipation than male subjects in the overall study population, and that the prevalence of abdominal fullness or bloating and of constipation were also higher in female than male subjects with normoglycemia. Logistical regression analyses revealed that female gender was associated with increased odds of at least one GI symptom and of lower GI symptoms. This is similar to what has previously been reported for the prevalence of GI symptoms and may reflect higher levels of psychological distress in females [3]. In our study, the prevalence of IBS was lower than in the Chinese community cohort of Sperber et al. (2.1% vs. 3.8%) [1]. This discrepancy may be due to differences in diagnostic criteria for IBS between the two studies, given that the DBSQ aligns with Rome II [5,12], while the global epidemiological study used Rome III criteria, which are slightly less restrictive in regard to the timing of symptoms. Park et al. [21] and Yao et al. [22] reported a slightly lower prevalence of IBS using Rome II compared to Rome III criteria. The study of Sperber et al. [1] showed a substantially lower prevalence of IBS when assessed by Rome IV criteria, which require the presence of abdominal pain rather than just discomfort, when compared to Rome III, both globally and in the Chinese cohort.

The observed prevalence of GI symptoms in individuals with diabetes in the present study was substantially lower than in an Australian community-based sample where symptoms were also ascertained using the DBSQ [5]. In the latter, 18.2% of patients reported upper GI symptoms (compared to 8.4% in our study) and 26.0% reported lower GI symptoms (compared to 12.1%) [5]. However, a key difference in this Australian study was that the DBSQ was completed by participants at home and returned by mail, while in our study, the DBSQ responses were recorded on the basis of face-to-face interviews. There is recent evidence that unsupervised questionnaire-based surveys yield a substantially higher GI symptom prevalence when compared to supervised surveys (e.g., internet questionnaires vs. direct interviews), possibly because some GI symptoms, such as fecal incontinence, are perceived as embarrassing [1]. Although we validated our approach in a small sample of inpatients with T2D and showed that the responses obtained by study interviewers showed good agreement to those elicited by a specialist endocrinologist, the possibility of the underestimation of GI symptoms cannot be excluded. In addition, cultural factors, social factors, ethnic and genetic differences and dietary factors may also contribute to the discrepancies in the prevalence of GI symptoms in different countries [23].

The observed rates of GI symptoms in individuals with diabetes were also substantially lower than those reported in previous studies in the Chinese population [14,24]. For example, Ko et al. [14] reported that up to 70% of outpatients with T2D had GI symptoms, while Huang et al. [24] reported that 62.2% of outpatients and inpatients with T2D had at least one GI symptom, with diarrhea and constipation being the most common, as in our study. However, our study was performed in a community setting; more than half of the individuals with diabetes were newly diagnosed, with relatively good glycemic control, and were therefore, less likely to have diabetes-related complications compared to patients attending tertiary centers. There were also substantial differences in the methodology used to assess GI symptoms between our study and previous reports which may, to some extent, have contributed to the differences in symptom prevalence. That we did not observe significant differences in the prevalence of GI symptoms between individuals with normoglycemia, prediabetes and diabetes, except that the symptoms of IBS were more prevalent in those with prediabetes, suggests that the occurrence of GI symptoms in general is not an early feature of hyperglycemia.

Several studies have explored the association of glycemic control with GI symptoms in diabetic patients [14,25,26,27]; however, the outcomes were inconsistent. In the present study, we found that neither FPG nor HbA1c were associated with the presence of GI symptoms, probably because glycemic control was good in the majority of participants. In our study, we observed an interaction between age (≥65 years vs. <65 years) and the presence of diabetes in relation to the prevalence of at least one GI symptom, and of constipation, which occurred in 34.8% and 17.4% of subjects with diabetes aged ≥ 65 years, respectively. The substantial burden of GI symptoms among elderly patients with diabetes necessitates specific consideration of the management of this subset of patients.

The occurrence of GI symptoms in patients with diabetes may represent an adverse effect of glucose-lowering medications, such as metformin, acarbose and glucagon-like peptide-1 receptor agonists [28]. For example, Bytzer et al. [29] reported that diarrhea and fecal incontinence were related to the use of metformin. However, the relationship between specific drugs and GI symptoms was not explored in our study due to limited data on the specific hypoglycemic agents used by patients with an established diagnosis of diabetes. However, our sensitivity analysis, excluding people with diabetes who were taking oral glucose-lowering drugs, with or without insulin, or had missing information regarding glucose-lowering therapies, showed that the outcomes were essentially unchanged.

### Strengths and Limitations

The current study represents the largest survey focusing specifically on the prevalence of GI symptoms in Chinese community-dwelling adults with and without diabetes, and the glycemic status of the majority of participants was categorized on the basis of an oral glucose tolerance test. Importantly, GI symptoms were assessed using a validated instrument, which was designed specifically for use in the context of diabetes. However, a number of limitations should be noted in interpreting our observations. First, since the population in our study was limited to Han Chinese, the results may not be representative of the entire Chinese population, including those of different ethnic backgrounds. Second, participants with a history of GI disease were excluded from our study, which may have contributed to the relatively low overall incidence of GI symptoms, although this minimized the potential for confounding in assessing the relationship between GI symptoms and diabetes. Third, patients with diabetes in the current study were in general well-controlled and uncomplicated, which might not be representative of the entire diabetic population. Fourth, given the lack of data on autoantibodies related to diabetes in the present study, the type of diabetes (i.e., type 1 vs. 2) could not be ascertained in the present study. Fifth, information on glucose-lowering therapies in subjects with diabetes was limited; the specific types of antidiabetic drugs or insulin therapy were not obtained in this study, although our sensitivity and subgroup analyses indicate that this did not influence the conclusions of our study. Sixth, GI symptoms might be a common feature in many rheumatic diseases, but information regarding such diseases were not collected for participants in the present study; however, the population of this study is derived from the general population and is relatively healthy. Seventh, previous studies have demonstrated that GI symptoms impact negatively on health-related quality of life and can be influenced by psychological function [1,30]; however, neither was evaluated in the current study. Finally, the use of questionnaires is inherently associated with the potential for recall bias.

## 5. Conclusions

In conclusion, we have demonstrated that GI symptoms are common in Chinese community-dwelling adults with and without diabetes, particularly in females, and that lower GI symptoms are the most common. Individuals with diabetes aged ≥ 65 years are particularly likely to have GI symptoms. These observations should alert clinicians to enquire specifically regarding GI symptoms in their clinical practice, and also highlight the need to document GI symptoms using validated questionnaires in clinical trials, in order to gauge their true prevalence.

## Figures and Tables

**Figure 1 nutrients-14-03506-f001:**
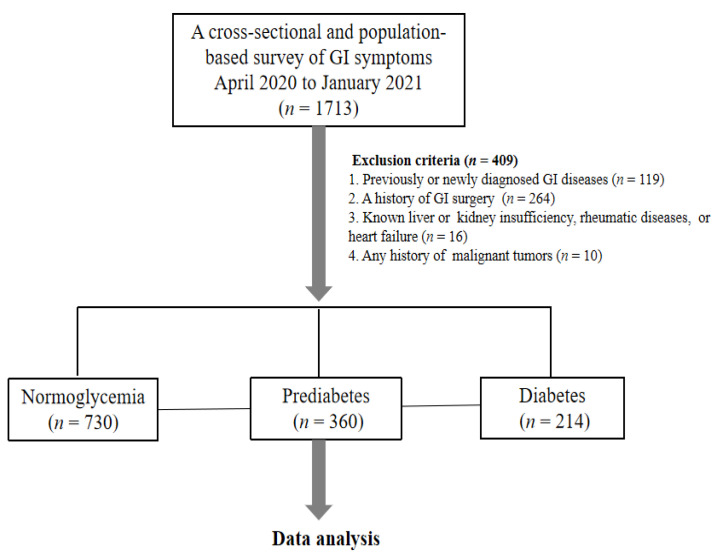
Flowchart of the study.

**Table 1 nutrients-14-03506-t001:** Characteristics of the study participants.

Characteristics	Normoglycemia (*n* = 730)	Prediabetes (*n* = 360)	Diabetes (*n* = 214)	*p*
Female, *n* (%)	460 (63.0%)	225 (62.5%)	121 (56.5%)	0.22
Age (y)	52.1 ± 9.9	55.7 ± 8.4 **	56.9 ± 8.8 **	<0.001
BMI (kg/m^2^)	25.0 ± 11.1	26.2 ± 10.4	26.2 ± 3.7	0.09
Waist circumference (cm)	83.9 ± 9.8	87.3 ± 9.3 **	89.1 ± 9.4 **	0.001
SBP (mmHg)	130.7 ± 18.4	137.7 ± 17.9 **	141.8 ± 20.3 **	<0.001
DBP (mmHg)	82.7 ± 12.3	86.4 ± 11.4 **	85.3 ± 14.1 *	<0.001
Heart rate (beats/min)	71.4 ± 11.0	72.5 ± 10.9	74.7 ± 11.4 **	0.001
FPG (mmol/L)	5.2 ± 0.4	5.6 ± 0.5 **	7.3 ± 1.8 **	<0.001
2hPG (mmol/L) ^a^	6.2 ± 1.0	8.8 ± 1.2 **	12.8 ± 2.5 **	<0.001
HbA1c (%)	5.5 ± 0.5	5.7 ± 0.5 **	6.8 ± 1.3 **	<0.001
Cr (umol/L)	56.6 ± 12.9	55.8 ± 13.7	57.2 ± 16.1	0.49
TC (mmol/L)	4.6 ± 0.8	4.7 ± 0.9 *	4.8 ± 0.9 **	0.002
TG (mmol/L)	1.2 (0.9, 1.7)	1.6 (1.1, 2.1) **	1.5 (1.1, 2.1) **	<0.001
HDL (mmol/L)	1.4 ± 0.2	1.4 ± 0.3	1.4 ± 0.3	0.46
LDL (mmol/L)	2.5 ± 0.6	2.6 ± 0.6	2.6 ± 0.7	0.07
Current Smoking, *n* (%)	128 (17.5%)	51 (14.2%)	35 (16.4%)	0.37
Alcohol intake				0.48
None, *n* (%)	537 (73.6%)	247 (68.6%)	149 (69.6%)	
Less than 10 g per week, *n* (%)	66 (9.0%)	36 (10.0%)	17 (7.9%)	
10–100 g per week, *n* (%)	77 (10.5%)	46 (12.8%)	31 (14.5%)	
100–300 g per week, *n* (%)	16 (2.2%)	15 (4.2%)	8 (3.7%)	
More than 300 g per week, *n* (%)	34 (4.7%)	16 (4.4%)	9 (4.2%)	
Newly diagnosed diabetes	NA	NA	123 (57.5%)	
Previously diagnosed diabetes	NA	NA	91 (42.5%)	
Duration of known diabetes (years) ^b^	NA	NA	6.5 ± 5.4	
Glucose-lowering therapies ^c^	NA	NA		
Oral hypoglycemic agents only, *n* (%)	NA	NA	46 (50.5%)	
Insulin ± oral hypoglycemic agents, *n* (%)	NA	NA	12 (13.2%)	
Lifestyle measures only, *n* (%)	NA	NA	18 (19.8%)	
Unclassified, *n* (%)	NA	NA	15 (16.5%)	
Diabetic complications				
Albuminuria ^d^	NA	NA	71 (34.9%)	
Diabetic retinopathy ^e^	NA	NA	22 (10.8%)	

Abbreviation: BMI, body mass index; SBP, systolic blood pressure; DBP, diastolic blood pressure; FPG, fasting plasma glucose; HbA1c, glycated hemoglobin; Cr, creatinine; TC, total cholesterol; TG, triglyceride; HDL, high-density lipoprotein; LDL, low-density lipoprotein. Data are presented as means ± SD or median (25th percentile, 75th percentile) or number (%). ^a^ Participants without 2hPG data (*n* = 112). ^b^ The duration of known diabetes in participants who were previously diagnosed with diabetes. ^c^ The glucose-lowing therapies in participants who were previously diagnosed with diabetes. ^d^ Participants without albuminuria data (*n* = 11). ^e^ Participants without retinopathy data (*n* = 12). Continuous variables across groups were compared using one-way analysis of variance or non-parametric Kruskal–Wallis test. Categorical variables across groups were compared using Chi-squared tests. * *p*
*<* 0.05 and ** *p*
*<* 0.001, compared to subjects with normoglycemia.

**Table 2 nutrients-14-03506-t002:** Prevalence of gastrointestinal symptoms in participants with normoglycemia, prediabetes and diabetes.

	Total (*n* = 1304)	Normoglycemia (*n* = 730)	Prediabetes (*n* = 360)	Diabetes (*n* = 214)	*p*
At least one GI symptom, *n* (%)	243 (18.6%)	129 (17.7%)	71 (19.7%)	43 (20.1%)	0.59
Abdominal pain, *n* (%)	15 (1.2%)	9 (1.2%)	4 (1.1%)	2 (0.9%)	1.00
Irritable bowel syndrome, *n* (%)	28 (2.1%)	10 (1.4%)	14 (3.9%) *	4 (1.9%)	0.025
Ulcer-like dyspepsia, *n* (%)	15 (1.2%)	9 (1.2%)	4 (1.1%)	2 (0.9%)	1.00
Early satiety, *n* (%)	29 (2.2%)	14 (1.9%)	10 (2.8%)	5 (2.3%)	0.66
Postprandial fullness, *n* (%)	23 (1.8%)	13 (1.8%)	5 (1.4%)	5 (2.3%)	0.71
Nausea, *n* (%)	7 (0.5%)	4 (0.5%)	3 (0.8%)	0 (0.0%)	0.44
Retching, *n* (%)	2 (0.2%)	1 (0.1%)	1 (0.3%)	0 (0.0%)	0.69
Vomiting, *n* (%)	2 (0.2%)	1 (0.1%)	1 (0.3%)	0 (0.0%)	0.69
Loss of appetite, *n* (%)	12 (0.9%)	8 (1.1%)	2 (0.6%)	2 (0.9%)	0.72
Abdominal fullness or bloating, *n* (%)	37 (2.8%)	21 (2.9%)	11 (3.1%)	5 (2.3%)	0.89
Gastroesophageal reflux symptoms, *n* (%)	38 (2.9%)	22 (3.0%)	8 (2.2%)	8 (3.7%)	0.56
Dysphagia, *n* (%)	2 (0.2%)	1 (0.1%)	1 (0.3%)	0 (0.0%)	0.69
Diarrhea, *n* (%)	79 (6.1%)	43 (5.9%)	23 (6.4%)	13 (6.1%)	0.95
Constipation, *n* (%)	72 (5.5%)	38 (5.2%)	22 (6.1%)	12 (5.6%)	0.83
Fecal incontinence, *n* (%)	9 (0.7%)	7 (1.0%)	1 (0.3%)	1 (0.5%)	0.53
Symptom complex					
Upper GI symptoms €, *n* (%)	102 (7.8%)	56 (7.7%)	28 (7.8%)	18 (8.4%)	0.94
Lower GI symptoms ∳, *n* (%)	153 (11.7%)	83 (11.4%)	44 (12.2%)	26 (12.1%)	0.89

Data are presented as number (%). €: A combined prevalence of early satiety, postprandial fullness, nausea, retching, vomiting, loss of appetite, abdominal fullness or bloating, gastroesophageal reflux symptoms and dysphagia. ∳: A combined prevalence of diarrhea, constipation and fecal incontinence. The prevalence of GI symptoms across groups was compared using Chi-squared tests. Logistic regression analyses were used to compare GI symptoms between the groups after adjusting for age and gender. * *p* < 0.05, compared to subjects with normoglycemia after adjusting for age and gender.

**Table 3 nutrients-14-03506-t003:** Prevalence of gastrointestinal symptoms in male and female participants.

	Groups		Total Male *n* = 498 Female *n* = 806	Normoglycemia Male *n* = 270 Female *n* = 460	Prediabetes Male *n* = 135 Female *n* = 225	Diabetes Male *n* = 93 Female *n* = 121	*p*
Symptoms							
At least one GI symptom, *n* (%)		Male	80 (16.1%)	38 (14.1%)	20 (14.8%)	22 (23.7%)	0.09
		Female	163 (20.2%)	91 (19.8%)	51 (22.7%)	21 (17.4%)	0.47
Abdominal pain, *n* (%)		Male	3 (0.6%)	1 (0.4%)	0 (0.0%)	2 (2.2%)	0.15
		Female	12 (1.5%)	8 (1.7%)	4 (1.8%)	0 (0.0%)	0.39
Irritable bowel syndrome, *n* (%)		Male	6 (1.2%)	2 (0.7%)	3 (2.2%)	1 (1.1%)	0.41
		Female	22 (2.7%)	8 (1.7%)	11 (4.9%)	3 (2.5%)	0.06
Ulcer-like dyspepsia, *n* (%)		Male	3 (0.6%)	1 (0.4%)	1 (0.7%)	1 (1.1%)	0.76
		Female	12 (1.5%)	8 (1.7%)	3 (1.3%)	1 (0.8%)	0.92
Early satiety, *n* (%)		Male	5 (1.0%)	2 (0.7%)	2 (1.5%)	1 (1.1%)	0.84
		Female	**24 (3.0%)** ** ^¶^ **	12 (2.6%)	8 (3.6%)	4 (3.3%)	0.77
Postprandial fullness, *n* (%)		Male	6 (1.2%)	4 (1.5%)	0 (0.0%)	2 (2.2%)	0.22
		Female	17 (2.1%)	9 (2.0%)	5 (2.2%)	3 (2.5%)	0.84
Nausea, *n* (%)		Male	3 (0.6%)	2 (0.7%)	1 (0.7%)	0 (0.0%)	1.00
		Female	4 (0.5%)	2 (0.4%)	2 (0.9%)	0 (0.0%)	0.63
Retching, *n* (%)		Male	1 (0.2%)	1 (0.4%)	0 (0.0%)	0 (0.0%)	1.00
		Female	1 (0.1%)	0 (0.0%)	1 (0.4%)	0 (0.0%)	0.43
Vomiting, *n* (%)		Male	0 (0.0%)	0 (0.0%)	0 (0.0%)	0 (0.0%)	/
		Female	2 (0.2%)	1 (0.2%)	1 (0.4%)	0 (0.0%)	0.68
Loss of appetite, *n* (%)		Male	1 (0.2%)	1 (0.4%)	0 (0.0%)	0 (0.0%)	1.00
		Female	**11 (1.4%)** ** ^¶^ **	7 (1.5%)	2 (0.9%)	2 (1.7%)	0.76
Abdominal fullness or bloating, *n* (%)		Male	9 (1.8%)	2 (0.7%)	4 (3.0%)	3 (3.2%)	0.09
		Female	28 (3.5%)	**19 (4.1%) ^#^**	7 (3.1%)	2 (1.7%)	0.39
Gastroesophageal reflux symptoms, *n* (%)		Male	16 (3.2%)	10 (3.7%)	3 (2.2%)	3 (3.2%)	0.79
		Female	22 (2.7%)	12 (2.6%)	5 (2.2%)	5 (4.1%)	0.57
Dysphagia, *n* (%)		Male	0 (0.0%)	0 (0.0%)	0 (0.0%)	0 (0.0%)	/
		Female	2 (0.2%)	1 (0.2%)	1 (0.4%)	0 (0.0%)	0.68
Diarrhea, *n* (%)		Male	32 (6.4%)	16 (5.9%)	8 (5.9%)	8 (8.6%)	0.64
		Female	47 (5.8%)	27 (5.9%)	15 (6.7%)	5 (4.1%)	0.63
Constipation, *n* (%)		Male	16 (3.2%)	6 (2.2%)	5 (3.7%)	5 (5.4%)	0.27
		Female	**56 (6.9%) ^#^**	**32 (7.0%) ^#^**	17 (7.6%)	7 (5.8%)	0.83
Fecal incontinence, *n* (%)		Male	1 (0.2%)	1 (0.4%)	0 (0.0%)	0 (0.0%)	1.00
		Female	8 (1.0%)	6 (1.3%)	1 (0.4%)	1 (0.8%)	0.78
Symptom complex							
Upper GI symptoms €, *n* (%)		Male	35 (7.0%)	17 (6.3%)	10 (7.4)	8(8.6%)	0.74
		Female	67 (8.3%)	39 (8.5%)	18 (8.0%)	10 (8.3%)	0.98
Lower GI symptoms ∳, *n* (%)		Male	48 (9.6%)	23 (8.5%)	12 (8.9%)	13 (14.0%)	0.29
		Female	105 (13.0%)	60 (13.0%)	32 (14.2%)	13 (10.7%)	0.66

Data are presented as number (%). €: A combined prevalence of early satiety, postprandial fullness, nausea, retching, vomiting, loss of appetite, abdominal fullness or bloating, gastroesophageal reflux symptoms and dysphagia. ∳: A combined prevalence of diarrhea, constipation and fecal incontinence. The prevalence of GI symptoms across groups was compared using Chi-squared tests. **^¶^**
*p*
*<* 0.05 and # *p*
*<* 0.01, compared to male subjects within each group. Bold values indicate a statistically significant difference between male and female groups.

**Table 4 nutrients-14-03506-t004:** Prevalence of gastrointestinal symptoms in participants <65 and ≥65 years.

	Groups		Total Age < 65 *n* = 1126 Age ≥ 65 *n* = 178	Normoglycemia Age < 65 *n* = 657 Age ≥ 65 *n* = 73	Prediabetes Age < 65 *n* = 301 Age ≥ 65 *n* = 59	Diabetes Age < 65 *n* = 168 Age ≥ 65 *n* = 46	*p*
Symptoms							
At least one GI symptom, *n* (%)		Age < 65	204 (18.1%)	119 (18.1%)	58 (19.3%)	27 (16.1%)	0.69
		Age ≥ 65	39 (21.9%)	10 (13.7%)	13 (22.0%)	**16 (34.8%) *** ** ^,⟂^ **	**0.03**
Abdominal pain, *n* (%)		Age < 65	14 (1.2%)	9 (1.4%)	3 (1.0%)	2 (1.2%)	0.93
		Age ≥ 65	1 (0.6%)	0 (0.0%)	1 (1.7%)	0 (0.0%)	0.59
Irritable bowel syndrome, *n* (%)		Age < 65	23 (2.0%)	10 (1.5%)	11 (3.7%)	2 (1.2%)	0.07
		Age ≥ 65	5 (2.8%)	0 (0.0%)	3(5.1%)	2 (4.3%)	0.11
Ulcer-like dyspepsia, *n* (%)		Age < 65	12 (1.1%)	9 (1.4%)	2 (0.7%)	1 (0.6%)	0.65
		Age ≥ 65	3 (1.7%)	0 (0.0%)	2 (3.4%)	1 (2.2%)	0.35
Early satiety, *n* (%)		Age < 65	25 (2.2%)	13 (2.0%)	8 (2.7%)	4 (2.4%)	0.79
		Age ≥ 65	4 (2.2%)	1 (1.4%)	2 (3.4%)	1 (2.2%)	0.82
Postprandial fullness, *n* (%)		Age < 65	21 (1.9%)	13 (2.0%)	4 (1.3%)	4 (2.4%)	0.68
		Age ≥ 65	2 (1.1%)	0 (0.0%)	1 (1.7%)	1 (2.2%)	0.51
Nausea, *n* (%)		Age < 65	6 (0.5%)	3 (0.5%)	3 (1.0%)	0 (0.0%)	0.50
		Age ≥ 65	1 (0.6%)	1 (1.4%)	0 (0.0%)	0 (0.0%)	1.00
Retching, *n* (%)		Age < 65	2 (0.2%)	1 (0.2%)	1 (0.3%)	0 (0.0%)	0.66
		Age ≥ 65	0 (0.0%)	0 (0.0%)	0 (0.0%)	0 (0.0%)	/
Vomiting, *n* (%)		Age < 65	2 (0.2%)	1 (0.2%)	1 (0.3%)	0 (0.0%)	0.66
		Age ≥ 65	0 (0.0%)	0 (0.0%)	0 (0.0%)	0 (0.0%)	/
Loss of appetite, *n* (%)		Age < 65	10 (0.9%)	7 (1.1%)	1 (0.3%)	2 (1.2%)	0.55
		Age ≥ 65	2 (1.1%)	1 (1.4%)	1 (1.7%)	0 (0.0%)	1.00
Abdominal fullness or bloating, *n* (%)		Age < 65	32 (2.8%)	20 (3.0%)	8 (2.7%)	4 (2.4%)	0.88
		Age ≥ 65	5 (2.8%)	1 (1.4%)	3 (5.1%)	1 (2.2%)	0.53
Gastroesophageal reflux symptoms, *n* (%)		Age < 65	31 (2.8%)	18 (2.7%)	8 (2.7%)	5 (3.0%)	0.98
		Age ≥ 65	7 (3.9%)	4 (5.5%)	0 (0.0%)	3 (6.5%)	0.11
Dysphagia, *n* (%)		Age < 65	2 (0.2%)	1 (0.2%)	1 (0.3%)	0 (0.0%)	0.66
		Age ≥ 65	0 (0.0%)	0 (0.0%)	0 (0.0%)	0 (0.0%)	/
Diarrhea, *n* (%)		Age < 65	70 (6.2%)	41 (6.2%)	18 (6.0%)	11 (6.5%)	0.97
		Age ≥ 65	9 (5.1%)	2 (2.7%)	5 (8.5%)	2 (4.3%)	0.35
Constipation, *n* (%)		Age < 65	58 (5.2%)	36 (5.5%)	18 (6.0%)	4 (2.4%)	0.20
		Age ≥ 65	14 (7.9%)	2 (2.7%)	4 (6.8%)	**8 (17.4%) *** ** ^,⟂^ **	**0.019**
Fecal incontinence, *n* (%)		Age < 65	7 (0.6%)	5 (0.8%)	1 (0.3%)	1 (0.6%)	0.87
		Age ≥ 65	2 (1.1%)	2 (2.7%)	0 (0.0%)	0 (0.0%)	0.34
Symptom complex							
Upper GI symptoms €, *n* (%)		Age < 65	86 (7.6%)	49 (7.5%)	24 (8.0%)	13 (7.7%)	0.96
		Age ≥ 65	16 (9.0%)	7 (9.6%)	4 (6.8%)	5 (10.9%)	0.75
Lower GI symptoms ∳, *n* (%)		Age < 65	129 (11.5%)	77 (11.7%)	36 (12.0%)	16 (9.5%)	0.69
		Age ≥ 65	24 (13.5%)	6 (8.2%)	8 (13.6%)	**10 (21.7%)** ** ^∓^ **	0.11

Data are presented as number (%). €: A combined prevalence of early satiety, postprandial fullness, nausea, retching, vomiting, loss of appetite, abdominal fullness or bloating, gastroesophageal reflux symptoms and dysphagia. ∳: A combined prevalence of diarrhea, constipation and fecal incontinence. The prevalence of GI symptoms across groups was compared using Chi-squared tests. Logistic regression analyses were used to compare GI symptoms between the groups after adjusting for age and gender. * *p* < 0.05, compared to participants with normoglycemia after adjusting for age and gender. ∓ *p* < 0.05 and ⟂ *p* < 0.01, compared to subjects who were less than 65 years old within each group. Bold values indicate a statistically significant difference between Age < 65 and Age ≥ 65.

**Table 5 nutrients-14-03506-t005:** Factors associated with the presence of GI symptoms.

	At least one GI Symptom		Abdominal Pain		Irritable Bowel Syndrome		Ulcer-like Dyspepsia		Upper GI Symptoms		Lower GI Symptoms	
	OR (95% CI)	*p*	OR (95% CI)	*p*	OR (95% CI)	*p*	OR (95% CI)	*p*	OR (95% CI)	*p*	OR (95% CI)	*p*
Gender												
Male	1.0		1.0		1.0		1.0		1.0		1.0	
Female	1.36 (1.01, 1.84)	0.04	2.51 (0.69, 9.07)	0.16	2.39 (0.95, 5.99)	0.06	2.63 (0.72, 9.69)	0.15	1.21 (0.79, 1.87)	0.38	1.44 (1.003, 2.08)	0.048
Age (years)	1.04 (1.02, 1.05)	<0.001	1.05 (0.99, 1.12)	0.11	1.03 (0.99, 1.08)	0.14	1.05 (0.98, 1.11)	0.15	1.04 (1.01, 1.06)	0.004	1.03 (1.01, 1.05)	0.005
BMI (kg/m^2^)	0.99 (0.97, 1.02)	0.49	1.00 (0.95, 1.05)	0.97	1.01 (0.98, 1.03)	0.62	0.99 (0.87, 1.13)	0.87	0.97 (0.91, 1.03)	0.35	0.99 (0.97, 1.02)	0.73
FPG (mmol/L)	1.04 (0.87, 1.24)	0.74	0.51 (0.20, 1.26)	0.14	0.65 (0.37, 1.14)	0.13	0.52 (0.23, 1.16)	0.11	1.12 (0.89, 1.42)	0.33	1.04 (0.85, 1.28)	0.69
HbA1c (%)	0.87 (0.67, 1.13)	0.33	1.46 (0.59, 3.58)	0.41	1.48 (0.81, 2.71)	0.20	1.73 (0.84, 3.59)	0.14	0.78 (0.54, 1.14)	0.20	0.92 (0.68, 1.25)	0.59

Abbreviation: BMI, body mass index; FPG, fasting plasma glucose; HbA1c, glycated hemoglobin. Multivariate logistic regression analyses were used to determine factors associated with GI symptoms (the model including age, gender, BMI, FPG and HbA1c).

## Data Availability

The datasets generated and/or analyzed during the current study are not publicly available, but are available from the corresponding author on reasonable request.

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
