# Peer review of "Prevalence of Gastrointestinal Symptoms in Chinese Community-Dwelling Adults with and without Diabetes"

_nutrients, 2022, doi:10.3390/nu14173506_

Round 1

Reviewer 1 Report

This prospective study of Chinese community-dwelling adults with or without diabetes aims to explore risk factors for GI symptoms using the Diabetes Bowel Symptom Questionnaire. Researchers randomly enrolled a subset of participants and compared GI symptoms incidence between a control group (normoglycemia) and two other groups (prediabetes and diabetes).

The total number of patients is significative (n=1,304) and well-distributed (control group=730, prediabetes=360 and diabetes=214).

The manuscript is well-written, and the study design is appropriate.

However, please find below the comments:

Major comments:

-          The study is well-designed, and the aim of this paper presented at the end of the introduction is attractive. Indeed, gastrointestinal symptoms occur in people with diabetes, but the exact prevalence is often unknown, and the underlying causes of GI symptoms are often not well characterized.

-          However, the findings of this paper are too general “GI symptoms are “common” in Chinese community-dwelling adults with or without diabetes” (conclusions of this paper), making the manuscript not very attractive and innovative.

-          Moreover, they conclude that females and the elderly with diabetes are at increased risk of symptoms. These conclusions remains too poor.

       in Part Results and especially in Table 1 (very precise and well-presented), the most significant potential risk factors of GI symptoms in participants with prediabetes or controls are lacking, such as the type of diabetes treatment (insulin, insulin, and tablets, tablets only, diet only). Consequently this potential important risk factor cannot be statistically assessed.

-          The variable glycemic control (good, bad, medium) is a lacking.

-          The authors did not specify if the patients who have type 2 diabetes or type 1 diabetes

-          Tables 3 and 4 should be replaced with a chart (columns).

-          Table 5 should present age- and sex-adjusted results.

-          The discussion is well-conducted and interesting, but the main findings of this study are too foregone.

Minor comments:

- the flow diagram of the study placed in supplementary data should be placed within the manuscript.

Reviewer 2 Report

Many questions arise when reading. it should be emphasized, However, that in the last part of the discussion there is the so-called criticism of the method. I propose to extract it from the discussion in a separate paragraph or in a separate section. it shows the authors' awareness of the limitations of the data they present. I am aware that the authors do not have a lot of relevant data that could influence the results, such as types of antidiabetic drugs, type of insulin therapy. The division of patients with diabetes according to its duration would undoubtedly enrich knowledge about the occurrence of GI symptoms. In addition, we do not know whether the patients suffered from DM1 or 2, which is also important and the text should contain, for example, information that the patients were not divided for this reason. Moreover, other concomitant diseases or their absence that could influence the GI symptoms, such as heart failure, cardiovascular diseases, rheumatic diseases, should also be given. if they did not occur, enter it as the exclusion criterion.

Round 2

Reviewer 1 Report

The authors have correctly responded to my comments and revised the manuscript accordingly. Now the paper can be published.